# Investigation of the knowledge, attitudes, and perceptions regarding the utilization of rosemary among the population in Jordan

Samar Thiab[1]*, Razan I. Nassar[2], Saif Alislam Alamleh[3], Abdullah Aboqubo[3], Abdullah Aljebori[3]

1 Associate Professor in Pharmaceutical Analysis, Department of Pharmaceutical Chemistry and Pharmacognosy, Faculty of Pharmacy, Applied Science Private University, Amman, Jordan, 2 Department of Clinical Pharmacy and Therapeutics, Faculty of Pharmacy, Applied Science Private University, Amman, Jordan, 3 Faculty of Pharmacy, Applied Science Private University, Amman, Jordan

* S_Thiab@asu.edu.jo

**Data Availability Statement:** All data is available within the manuscript.

**Funding:** The author(s) received no specific funding for this work.

## Abstract

### Background

Rosemary (*Rosmarinus officinalis*) has a rich historical use for various reasons due to its beneficial characteristics including stimulating hair growth, providing antioxidant and anti-bacterial effects, and functioning as a skin conditioner and fragrance enhancer. The plant is cultivated in Jordan and using it is gaining popularity among the population.

### Aim of the study

To assess the knowledge, attitude and perception of rosemary use among Jordanians.

### Materials and methods

A cross-sectional study was conducted targeting at least 385 participants via social media platforms. Face and content validity of the questionnaire was performed by independent researchers. The questionnaire was divided into three main sections including the demographics section, the knowledge section and the attitude toward rosemary use. Statistical analysis was conducted using SPSS including descriptive statistics, chi-square tests, and multiple linear regression.

### Results

The study's participants (n = 407) had a mean age of 30.98 years old (SD = 12.76). The mean knowledge score for the participants ranged from -5 to 7, with a mean of 2.96 (SD = 2.61), with younger participants exhibiting higher knowledge scores regarding rosemary. Approximately half of the participants (48.2%) reported using rosemary for medical purposes. About three-quarters reported using rosemary oil, but a higher percentage reported using the leaves. Around three quarters administered it orally or applied it topically. Most participants obtained rosemary by growing it at home or from herbalists. Family and friends were the main influencers for rosemary users. Most of the study participants strongly

**Competing interests:** The authors have declared that no competing interests exist

agreed/agreed that plants possess a healing power (94.4%). The most reported reason for using rosemary was improving the hair condition (82.1%) followed by gastrointestinal problems (67.9%).

## Conclusions

The study highlights the widespread use of rosemary for different reasons through different methods of application. Addressing misconceptions and enhancing knowledge dissemination may allow informed decision-making and promote the safe and effective use of herbal therapies.

## Introduction

The use of herbs for medicinal purposes was found to be approximately 80% within Arab societies [1, 2]. For instance, in Egypt, around 37% of the population reported utilizing herbal medicines [3], whereas in Saudi Arabia, an even higher proportion, accounting for 73%, have reported themselves as users of herbal remedies [4]. Notably, the prevalence of herbal usage in Jordan emerges as the highest among Middle Eastern regions, standing at an impressive 80.2%, as evidenced by comparisons with studies conducted elsewhere in the region [5].

Among the commonly used herbs is rosemary (*Rosmarinus officinalis*). Rosemary is a fragrant, evergreen shrub with needle-like leaves. It belongs to the mint family, Lamiaceae. The leaf sizes, branch development patterns, and the colors of the flowers, which could be white, pink, purple, or blue differ for different rosemary varieties [6]. Additionally, the content of the plant's essential oils and oleoresins [7, 8] differ among the different rosemary varieties. In addition to the volatile constituents, rosemary was found to contain a variety of phytochemicals including flavonoids, polyphenols and terpenoids [9].

Rosemary's historical significance is far-reaching and diverse, it was used in Ancient Greece as a memory-enhancing agent and was utilized in Ancient Egypt for corpses embalming. Moreover, in 1235, Queen Isabella of Hungary used the infusion of rosemary in spirits of wine, known as "Hungary water," as a remedy for her paralyzed limbs [7, 10].

Recently, aside from the use of rosemary in food industries, it has several medical applications [6]. It was reported to be used for promoting hair growth, alleviating anxiety and stress, providing adjuvant support for breast cancer treatment, enhancing cognitive performance, alleviating constipation, managing dermatitis, and mitigating the challenges posed by rheumatic diseases [10–12]. In addition, rosemary also was found to have antioxidant and antibacterial activities and serves as a skin conditioner and fragrance enhancer [10, 13–15].

Rosemary is cultivated in Jordan making it easily accessible and available to the general population [16, 17]. It was reported in several studies that Jordanians use rosemary for various health conditions [18, 19]. The majority of the mentioned studies were exploring the use of herbal medicine in general and not focusing on a specific one.

This study aims to investigate how often people in Jordan use rosemary and its oil for health purposes, which was not conducted before. In addition to understand how much they know about the health benefits of rosemary. This research could provide valuable insights into the role of traditional herbal remedies in modern healthcare practices and potentially benefit the local community's well-being.

## Materials and methods

### Study design and participants

A cross-sectional study was conducted, after the development and validation (both face and content validity) of a self-administered electronic questionnaire to gather anonymous responses from participants. Eligible participants were any interested Jordanian citizens living in Jordan. The questionnaire was distributed using WhatsApp® and other social media platforms such as Facebook®, Twitter® and LinkedIn®. A snowball sampling method was employed. As initial potential participants were recruited, and then they were asked to refer other potential participants who met the study's inclusion and were willing to complete the survey. The process continued until the reached sample size was sufficient and representative to meet the study's objectives. The questionnaire's first page provided a detailed overview of the study, emphasizing voluntary participation and the confidential treatment of responses. Participants recruitment and data collection lasted for a month, starting on the 21st November, 2023 and ending on the 20th December, 2023.

### Questionnaire development

The questionnaire (S1 Appendix) was developed after reviewing validated surveys in literature [6, 7, 20–23] and was designed using the general principles of good survey design [24–26]. The questionnaire needed around 5 minutes to be completed and was administered in Arabic for the Jordanian population. The questionnaire consisted of three sections with the first section designed to collect data on the participants' socio-demographic characteristics (12 questions); the second section was designed to assess the knowledge (7 questions) about the medicinal uses of rosemary, and a question regarding their source of information regarding herbs; while the third section was prepared to assess the attitude and practice toward rosemary and its oil (16 questions). The survey's items were created based on a review of the literature and after consultation with experts in the field ensuring the comprehensive of items. 'Yes' and 'No' dichotomous questions were chosen for the knowledge section, which provide a clear binary response. This format was selected due to its ability to precisely target the specific knowledge area relevant to the current research, which may not be fully covered in the existing published literature [27].

### Validation and reliability

The initial version of the questionnaire underwent evaluation by members of the research team, who made revisions to improve its clarity and readability. Subsequently, a professional committee comprising clinical pharmacists and a statistician assessed the questionnaire's validity and reliability, confirming its suitability for the Jordanian population. Following this, the questionnaire was translated into Arabic and back-translated by two proficient bilingual academic staff members to ensure linguistic accuracy. The questions were formulated to avoid medical terminology, and the questionnaire underwent validation on a pilot sample consisting of 10 individuals from academic and non-academic backgrounds over the course of a month. This pilot study aimed to assess comprehension, clarity, readability, and overall acceptability of the survey, with adjustments to the questions made as necessary before its full implementation. Internal consistency reliability was tested by Cronbach's alpha coefficient, which resulted in a coefficient of 0.76.

### Sample size

Currently, the number of Jordanian citizens living in Jordan is 11.516 million [28]. Based on that, calculating the sample size using a margin of error of 5%, confidence level of 95%, and

response distribution of 50%, a minimum sample size of 385 is needed. The number of participants in this study was 407 [29].

## Statistical analyses

Statistical analysis was conducted using SPSS version 21.0 for Windows (SPSS, Inc., Chicago, IL). Descriptive statistics including percentages, means, and frequency distribution were calculated for each of the questions. The Pearson chi-square test was used to assess the relationship among gender and rosemary users. Statistical significance was defined as a p-value of 0.05. Multiple linear regression was conducted to screen for the independent variables affecting the rosemary knowledge scores. Initially, simple linear regression was conducted and any independent variable that had a p-value less than 0.25 was eligible to be entered into the multiple linear regression. Afterward, multiple linear regression was conducted, any variable with a p-value of 0.05 or less was considered statistically significant. All independent variables were selected after assessing their independence, confirming a tolerance value of <0.2, and a variance inflation factor (VIF) of <5 to approve the absence of multicollinearity.

A knowledge score was calculated using the seven items included in the third section, resulting in the computation of a score on a scale of -7 to 7. The scoring system allocated a value of +1 for each correct answer, 0 for an "I do not know" response, and -1 for an incorrect answer. No cut-off score value determined the acceptable level of knowledge. A good degree of knowledge was defined in this study as having at least 50% of the total right answers [30]. Additionally, a higher score in each category denotes greater understanding.

## Ethics approval

Ethical approval for this study was obtained from the Institutional Review Board Committee at the Faculty of Pharmacy, Applied Science Private University (Approval Number: 2023-PHA-37). Participants provided informed consent by indicating their acceptance to proceed with the study's survey. All study participants were adults, thus, obviating the need for guardianship.

## Results

A total of 409 potential participants opened the study's survey. Out of these, two participants chose the 'disagree' button, indicating their refusal to participate. As a result, 407 participants were included in the analysis, yielding a response rate of 99.51%.

The study's participants (n = 407) had a mean age of 30.98 (SD = 12.76). About 70.0% of the participants were females (n = 284), with half of the study participants being single (n = 204). The education level varied among participants ranging from those who did not complete their study (n = 5) to those who hold a postgraduate degree (n = 43). Most of the study participants were living in the center regions of Jordan (n = 325). Regarding the monthly income, 38.1% reported earning <250 Jordanian Dinars (JD), 25.8% from 250 to 500 JD, 21.1% from 501 to 750 JD, 5.7% from 751 to 1000 JD, and 9.3% more than 1000 JD. More than 60.0% of the study participants were insured, and about two-thirds were non-smokers (n = 270). With regards to participants' health, 14.3% reported having a chronic disease such as hypertension or diabetes (n = 58), and 20.6% reported taking medications (n = 84). The detailed demographic characteristics of the study participants are listed in Table 1.

Assessing participants' (n = 407) knowledge regarding rosemary revealed that the most correctly answered statement was "*Rosemary cannot be used as a condiment for food*" as 63.1% (n = 257) of the participants knew that it was written incorrectly. This was followed by the correctly written item "*Rosemary has antioxidant properties*", as 58.7% (n = 239) answered it

**Table 1. Demographic characteristics of the study's participants (n = 407).**

| Parameter | n (%) |
|---|---|
| **Gender** | |
| Male | 123 (30.2) |
| Female | 284 (69.8) |
| **Marital status** | |
| Single | 204 (50.1) |
| Married | 187 (45.9) |
| Divorced | 11 (2.7) |
| Widowed | 5 (1.2) |
| **Educational level** | |
| I did not complete my study | 5 (1.2) |
| High school certificate | 43 (10.6) |
| Diploma | 53 (13.0) |
| Bachelor's degree | 263 (64.6) |
| Postgraduate degree | 43 (10.6) |
| **Employment** | |
| Student | 167 (41) |
| Unemployed | 63 (15.5) |
| Academic | 40 (9.8) |
| Employee in the health sector | 30 (7.4) |
| Employee in a non-health sector | 44 (10.8) |
| Freelancers or business owner | 29 (7.1) |
| Retired | 13 (3.2) |
| Other | 21 (5.2) |
| **Place of residence** | |
| North regions (Irbid, Ajloun, Jerash, and Mafraq) | 45 (11.1) |
| Centre regions (Amman, Zarqa, Balqa, and Madaba) | 325 (79.9) |
| South regions (Karak, Tafilah, Maʻan, and Aqaba) | 37 (9.1) |
| **Monthly Income** | |
| <250 JD | 155 (38.1) |
| 250–500 JD | 105 (25.8) |
| 501–750 JD | 86 (21.1) |
| 751–1000 JD | 23 (5.7) |
| >1000 JD | 38 (9.3) |
| **Health Insurance Status** | |
| Insured | 252 (61.9) |
| Uninsured | 155 (38.1) |
| **Do you use/smoke tobacco products?** | |
| Yes | 137 (33.7) |
| No | 270 (66.3) |
| **Do you suffer from any chronic disease(s), such as hypertension, diabetes, etc.?** | |
| Yes | 58 (14.3) |
| No | 349 (85.7) |
| **Do you take any chronic medications (anti-hypertensive, diabetic drugs, etc. . . .)?** | |
| Yes | 84 (20.6) |
| No | 323 (79.4) |

**Table 2. Knowledge about the medicinal uses of rosemary among the study's participants (n = 407).**

| Parameter | True n (%) | I do not know n (%) | False n (%) |
|---|---|---|---|
| Rosemary has antioxidant properties | 239 (58.7) | 145 (35.6) | 23 (5.7) |
| Rosemary has antibacterial properties | 237 (58.2) | 149 (36.5) | 21 (5.2) |
| Rosemary cannot be used as a natural aroma* | 109 (26.8) | 121 (29.7) | 177 (43.5) |
| Rosemary has antifungal properties | 210 (51.6) | 166 (40.8) | 31 (7.6) |
| Rosemary has anti-inflammatory properties | 254 (62.4) | 119 (29.2) | 34 (8.4) |
| Rosemary cannot be used as a fragrance* | 76 (18.7) | 149 (36.6) | 182 (44.7) |
| Rosemary cannot be used as a condiment for food* | 58 (14.3) | 92 (22.6) | 257 (63.1) |

*Incorrect item

correctly with '*True*', and "*Rosemary has antibacterial properties*", as 58.2% (n = 237) identified that the statement is correct (Table 2). On a scale ranging from -7 (minimum) to 7 (maximum), the mean knowledge score for the participants ranged from -5 to 7, with a mean of 2.96 (SD = 2.61).

According to the multiple linear regression analysis (Table 3), the participant's age was the only statistically significant variable that affected the rosemary knowledge score (p-value = 0.005).

When study participants were asked whether they use rosemary for medical purposes, 48.2% (n = 196) responded with 'Yes', and 51.8% (n = 211) responded with 'No'. Among those

**Table 3. Assessment of factors affecting knowledge scores among rosemary users (n = 196).**

| Parameter | Knowledge score | | | |
|---|---|---|---|---|
| | Beta | P-value# | Beta | P-value$ |
| Age | -0.316 | <0.001^ | -0.217 | **0.005*** |
| Gender | | | | |
| • Male | Reference | | | |
| • Female | 0.137 | 0.506^ | 0.042 | 0.557 |
| Educational level | | | | |
| • < Bachelor's degree | Reference | | | |
| • ≥ Bachelor's degree | -0.017 | 0.809 | — | — |
| Do you have any chronic disease/s? | | | | |
| • No | Reference | | | |
| • Yes | -0.299 | <0.001^ | -0.248 | 0.062 |
| Do you take any chronic medication/s? | | | | |
| • No | Reference | | | |
| • Yes | -0.214 | 0.003^ | 0.078 | 0.536 |

# Using simple linear regression

$ Using multiple linear regression

^ Eligible for entry in multiple linear regression (significant at 0.25 significance level)

*Significant at 0.05 significance level

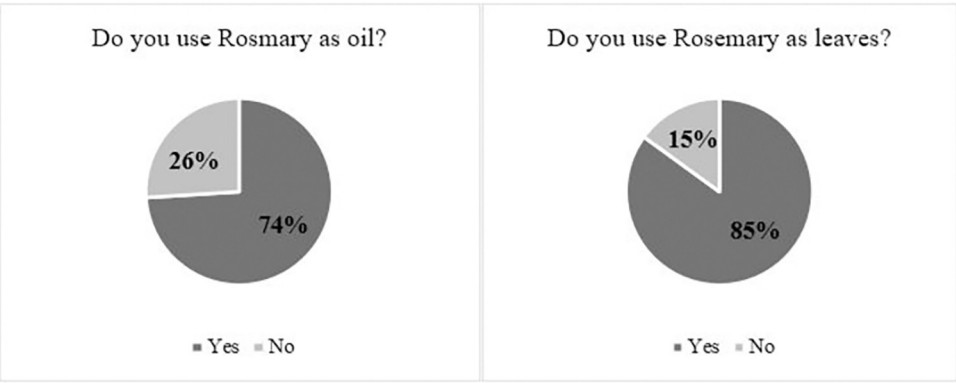

**Fig 1. The nature of used rosemary among the study participants (n = 196).**

who have previously used rosemary (n = 196), 74.0% reported using it in oil form (n = 145), while 26.0% did not use it as oil before (n = 51). Furthermore, 84.7% reported using rosemary leaves (n = 166), while 15.3% (n = 30) did not use rosemary leaves before (Fig 1).

About 55.0% of those who used rosemary previously reported using it as an inhalation (n = 108), and more than three-quarters (77.6%) administered it orally. Moreover, 70.9% (n = 139) applied it topically with gentle rubbing on a specific area (Fig 2).

Regarding the source of rosemary, 37.2% (n = 73) grow it at home, 34.7% (n = 68) buy it from herbalists, 19.9% buy it online (n = 39), 7.1% buy it from the vegetable market (n = 14), and 2 participants reported other sources. The participants reported being advised or influenced to use rosemary by family and friends (56.7%) as well as social media (20.4%). Most of the study participants (94.4%) strongly agreed/agreed that plants possess a healing power (Table 4).

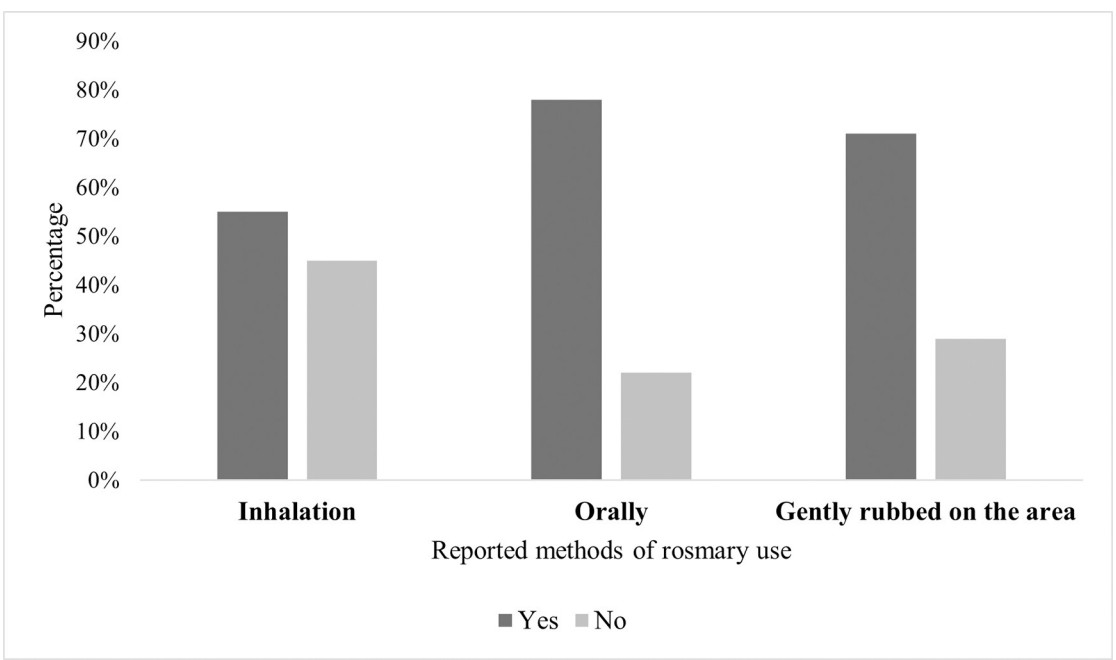

**Fig 2. Percentages of the methods of applying or using rosemary as reported by the study's participants (n = 196).**

**Table 4. Attitude and practice toward rosemary among the study's participants\*.**

| Parameter | n (%) |
|---|---|
| **Where did you get the rosemary/oil that you used for medical purposes?** | |
| I bought it from an herbalist | 68 (34.7) |
| I bought it from a vegetable market | 14 (7.1) |
| I grow it at home | 73 (37.2) |
| I have bought it online | 39 (19.9) |
| Others | 2 (1.0) |
| **Who advised you to use rosemary or its oil?** | |
| A doctor | 11 (5.6) |
| A pharmacist | 14 (7.1) |
| A nurse | 5 (2.6) |
| Family and friends | 111 (56.7) |
| I read about it on social media | 40 (20.4) |
| I heard about it on TV | 1 (0.5) |
| I read about it in medical journals | 7 (3.6) |
| I just decided to use it by myself | 7 (3.6) |
| **Do you believe in the healing power of plants?** | |
| Strongly Agree | 143 (73.0) |
| Agree | 42 (21.4) |
| Neutral | 9 (4.5) |
| Disagree | 2 (1) |
| Strongly Disagree | 0 (0) |

\*For participants who used rosemary before (n = 196).

Participants were asked whether they have used rosemary or its oil for several medical purposes (Fig 3). The most reported reason was hair condition (82.1%, n = 161), followed by gastrointestinal tract (GIT) problems such as spasms, ulcers, inflammation (67.9%, n = 133), stress or anxiety (66.8%, n = 131), memory enhancement (55.1%, n = 108), skin conditions (54.1%, n = 106), joint problems (40.8%, n = 80), and lastly for hypercholesterolemia (26.5%, n = 52).

A high percentage of the study participants (95.9%, n = 188) reported that rosemary or its oil contributed to relieving their health issues, and 4.1% (n = 8) reported the opposite.

## Discussion

This is the first study to focus on the use of one of the most common natural products available in Jordan, rosemary. The findings offer insights into the prevalence of rosemary use, methods of application, sources of acquisition, reasons for use, and perceived efficacy, alongside factors influencing knowledge, where age was found to be the only significant variable that affected the rosemary knowledge score.

Most of the study participants were females as it was noted in previous studies conducted in the United States, the United Kingdom, Australia and Germany [31–35] that they are more into herbal and alternative medicine when compared to men, and so they are more willing to participate in studies regarding these matters.

Younger participants were found to be more knowledgeable about rosemary properties and uses, this can be due to the fact that younger generation have more access to technological devices and online platforms and medical websites compared to older generation [36], which

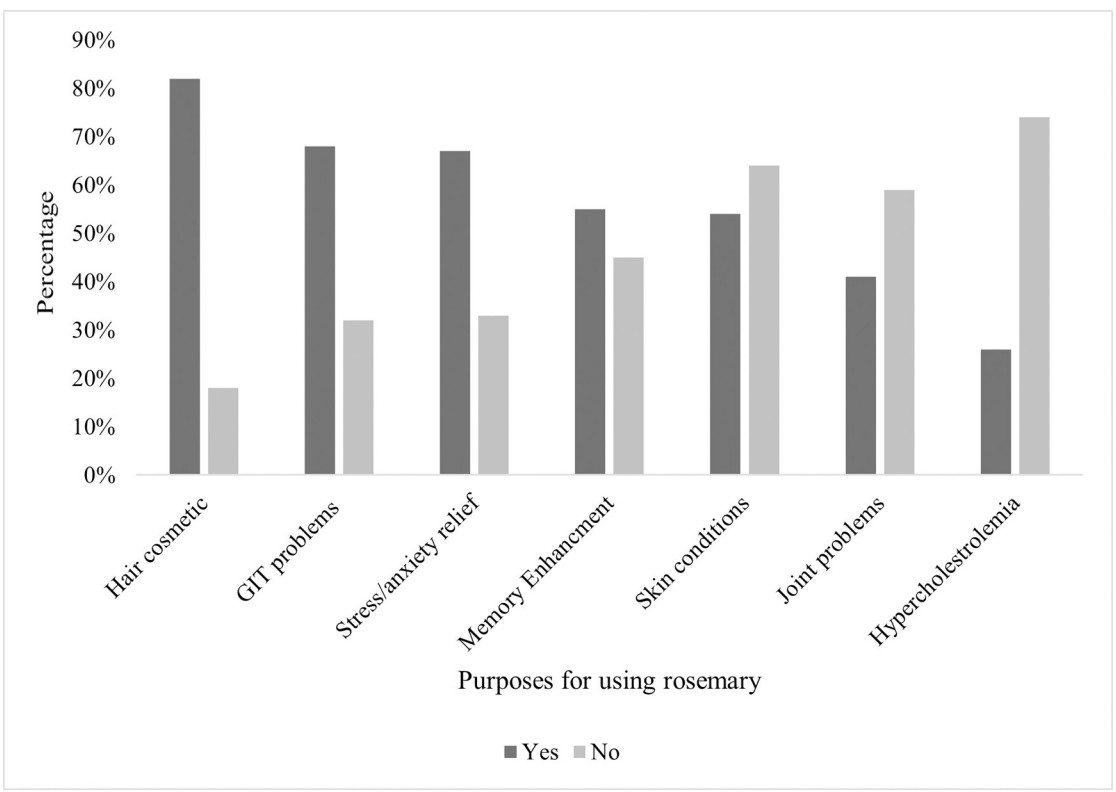

**Fig 3. Percentages of survey participants (n = 407) reporting the use of rosemary for several conditions.**

can help them obtain information easier and faster. Additionally, younger people are usually more curious than older people [37], and this could be another reason to why they search for more information about the herbs and medicine that they are about to use.

Around half of the participants reported the use of rosemary oil and leaves for medical purposes. The popularity of rosemary oil versus rosemary leaves for health benefits can vary depending on cultural practices, personal preferences, and specific health concerns. Rosemary oil, leaves, and berries were all reported to be used in folk medicine including in ancient Egypt, Hungary, London, and France [7]. Moreover, in a study conducted in Morocco targeting patients with kidney diseases revealed that that most used herbal medicine was *Rosmarinus officinalis* [38].

Since rosemary is a plant of Mediterranean origin [7], it can easily be cultivated by the general population in their gardens and homes, and as a result in this study, more than a third of the study participants were doing so. Family members and friends played a key role in influencing the study participants to use rosemary. Similar finding was obtained in other cross-sectional studies conducted in Jordan, where family and friends were found to have a great influence on one another [5, 39]. In Türkiye, friends were found to be the main source of information for herbal medicines users [21]. Additionally, in a survey conducted in central region of Saudi Arabia, the most common influences for using herbal medicine were found to be family and friends followed by the internet [20].

In the current study, the majority of the participants revealed that rosemary was efficient in relieving their health complaints. Several reasons were reported for the use of rosemary including hair condition, GIT problems, stress, anxiety, memory enhancement, skin conditions, joint problems, and hypercholesterolemia. This is consistent with the scientific evidence

regarding rosemary's biological activities. In a randomized comparative trial, rosemary oil was found to be efficient for the treatment of androgenetic alopecia [40], other studies also supported the beneficial effect of rosemary on hair [13, 41]. It is also well documented that rosemary has antinociceptive, antioxidant, antimicrobial, and anti-inflammatory properties [7, 9, 42, 43], furthermore that it is useful in alleviating different GI problems [11, 44]. Rosemary was also found to be beneficial in improving skin condition and promoting wound healing [45, 46]. It is reported that it has anxiolytic properties and were found to increase the level of brain-derived neurotrophic factor (BDNF) when rosemary tea was consumed for ten days by healthy human volunteers [47]. Moreover, rosemary was found to improve the human short-term image and numerical memory in a study involved secondary school students in Ukraine, where rosemary essential oil was sprayed onto one group and compared with the control group [48]. Similar memory effects were noted in college students in a study conducted in Korea using rosemary aromatic candles [49]. In addition, it was found that rosemary can improve cholesterol levels and glycaemia in mice [50]. Similar findings were noted in type-2 diabetic human patients, who took 3 grams of rosemary per day for four weeks [51].

This study has limitations. The recruitment of study participants was based on self-selection through social media which affects the generalizability of the findings due to the possibility of sampling bias. Another limitation is the representativeness of the sample to the population of Jordan as females actively participated in this survey are more than males and most of the participants were from the center of Jordan where the capital Amman is, however, this provides an opportunity for future studies to access the knowledge, attitude, perception and use of rosemary more extensively nationwide.

## Conclusion

The study offers valuable insights regarding knowledge, attitudes and perception, about rosemary use in Jordan. The results highlight how common it is to utilize rosemary for a variety of health-related objectives, and how effective people believe rosemary to be in alleviating health issues. Addressing misconceptions and enhancing knowledge dissemination may foster informed decision-making and promote safe and effective utilization of herbal therapies like rosemary in healthcare practices.

## Supporting information

**S1 Appendix. The questionnaire.**
(DOCX)

**S1 Graphical abstract.**
(TIF)

## Author Contributions

**Conceptualization:** Samar Thiab, Saif Alislam Alamleh, Abdullah Aboqubo, Abdullah Aljebori.

**Data curation:** Saif Alislam Alamleh, Abdullah Aboqubo, Abdullah Aljebori.

**Formal analysis:** Razan I. Nassar, Saif Alislam Alamleh, Abdullah Aboqubo, Abdullah Aljebori.

**Methodology:** Samar Thiab, Razan I. Nassar.

**Project administration:** Samar Thiab.

**Supervision:** Samar Thiab.

**Writing – original draft:** Samar Thiab, Razan I. Nassar, Saif Alislam Alamleh, Abdullah Aboqubo, Abdullah Aljebori.

**Writing – review & editing:** Samar Thiab, Razan I. Nassar, Saif Alislam Alamleh, Abdullah Aboqubo, Abdullah Aljebori.

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
