## [Decision Letter · Decision Letter 0]

27 May 2024

PONE-D-24-15316An Investigation into the Knowledge, Attitudes, and Perceptions (KAP) Regarding the Utilization of Rosemary and Rosemary Oil among the General Population in JordanPLOS ONE

Dear Dr. Thiab,

Thank you for submitting your manuscript to PLOS ONE. After careful consideration, we feel that it has merit but does not fully meet PLOS ONE’s publication criteria as it currently stands. Therefore, we invite you to submit a revised version of the manuscript that addresses the points raised during the review process.

Please submit your revised manuscript by Jul 11 2024 11:59PM. If you will need more time than this to complete your revisions, please reply to this message or contact the journal office at plosone@plos.org. Please include the following items when submitting your revised manuscript:A rebuttal letter that responds to each point raised by the academic editor and reviewer(s). You should upload this letter as a separate file labeled 'Response to Reviewers'.A marked-up copy of your manuscript that highlights changes made to the original version. You should upload this as a separate file labeled 'Revised Manuscript with Track Changes'.An unmarked version of your revised paper without tracked changes. You should upload this as a separate file labeled 'Manuscript'.If applicable, we recommend that you deposit your laboratory protocols in protocols.io to enhance the reproducibility of your results. Protocols.io assigns your protocol its own identifier (DOI) so that it can be cited independently in the future. For instructions see: https://journals.plos.org/plosone/s/submission-guidelines#loc-laboratory-protocols. Additionally, PLOS ONE offers an option for publishing peer-reviewed Lab Protocol articles, which describe protocols hosted on protocols.io. Read more information on sharing protocols at https://plos.org/protocols?utm_medium=editorial-email&utm_source=authorletters&utm_campaign=protocols.

We look forward to receiving your revised manuscript.

Kind regards,

Othman A. Alfuqaha, Ph.D.

Academic Editor

PLOS ONE

Journal Requirements:

2. In the online submission form, you indicated that [Data is available on request due to privacy or ethical restrictions.]. 

Additional Editor Comments:

Dear authors,

Thank you for your submission. After careful review, the reviewers have provided valuable feedback on your paper. It is strongly recommended that you carefully consider and address their comments in order to proceed in a positive direction. Additionally, I encourage you to take into account my own comments to further enhance your paper and strengthen its overall quality.

Editor Comments:

Title: Shorten the title to no more than 15 words for conciseness, and please remove the abbreviation.

Abstract: Your email is to validate questionnaire and assess the knowledge. Please added aims accordingly. Include details about your data collection methods such as whether you used random, snowball, or convenience sampling. Moreover, add the time of data collection. Construct validity why is not included in your study?

Introduction: References in Brackets. Add paragraph about scales used to measure knowledge and why the selected scale is important. Add the aim of validating a questionnaire. Add what the problem or useful of using Rosmarinus among Jordanians.

Method Section: Provide more information on how you conducted the online survey, ensuring it is clear how participants from Jordan were able to respond. How you reached 407 of participants. Is any outliers in incomplete surveys.

Questionnaire development: Where are the references regarding development? Consider adding construct validity measures such as correlations, KMO, Bartlett's test of sphericity, and total variation in addition to CVR ratios (content validity).

Typos: Carefully review and correct any typos present in the manuscript.

Discussion: Begin the discussion section with a focus on your main findings rather than repeating the study's aims.

International Comparisons: Compare your results with findings from international studies to highlight key differences.

References: Double-check and ensure the accuracy of your references.

We appreciate your commitment to revising your manuscript. Best of luck in this revision journey.

Sincerely,

Dr. Alfuqaha

Reviewers' comments:

Reviewer's Responses to Questions

**Comments to the Author**

1. Is the manuscript technically sound, and do the data support the conclusions?

Reviewer #1: Yes

Reviewer #2: Partly

2. Has the statistical analysis been performed appropriately and rigorously? 

Reviewer #1: Yes

Reviewer #2: Yes

3. Have the authors made all data underlying the findings in their manuscript fully available?

Reviewer #1: Yes

Reviewer #2: Yes

4. Is the manuscript presented in an intelligible fashion and written in standard English?

Reviewer #1: Yes

Reviewer #2: Yes

5. Review Comments to the Author

Reviewer #1: Reviewer comments

Abstract

• Write the unit for the age.

• 70. Should be written in percentage form

• The introduction writing sounds more literary than scientific. It needs to focus scientifically on the uses, active ingredients and how they proved therapeutic effect and significance.

Methodology:

• It is well-written but there is a redundancy in the sentence “to proceed with the study study's survey” needs to be edited.

• How the survey was disseminated to the public? You need to mention this in the methodology section

Results:

• It is a good practice to mention if there were incomplete responses or if there was any response that was excluded from the analysis.

• Define abbreviations at the first instance (e.g. JoD)

• Figure 2 and 3: define the y-axis title

• Few grammatical issues such as “its’”

• I think that some statements containing negation such as “Rosemary cannot be used as a natural aroma” were leading.

Reviewer #2: This study investigates the Jordanians’ attitude, practice and knowledge towards rosemary and rosemary oil. I suggest accepting it for publication but after amending it based on the following comments

Title

Although in the title the authors mentioned “Knowledge” first, they left everything about it in the M&M and Results to the end. I don’t agree to that because first you have to investigate about the knowledge then the attitude and the practice. Therefore, I suggest that the authors arrange the manuscript to talk first about the knowledge then the other components. In this way there will be synchronization between the title and the different parts of the manuscript. I strongly encourage sending the manuscript to English editing

Abstract:

1- The last sentence in the background has to be reformulated

2- In the Methods part, it is better to mention the sample number

3- In the method section, you have to arrange the sentence talking about the what this paper is assessing. You should start with knowledge then practice and attitude

4- In the methods, it would be important to mention which geographic parts of Jordan where covered

5- In the results part: add % for the last number of “More than half reported using it as inhalation (55.0%), whereas 77.6% administered it orally, and 70.9 applied it topically.”

Introduction

1- It would be more informative if you mention about the percentage of the population in geographic regions other than in the Arab world in the first paragraph. This would be useful to see the differences.

2- You need to add reference for this sentence “Rosemary's historical significance is far-reaching and diverse: from being revered in Ancient Greece as a memory-enhancing agent to gracing the burial rituals of Egyptian pharaohs while also being utilized in the embalming of corpses.”

3- I think the introduction needs to have few sentences more mentioning if similar study was conducted previously in jordan and in other countries or not. This should lead to the concluding sentence about the importance of this study, at the end of the Introduction section

M&M

1- This section is missing information about the geographic areas targeted, number of questionnaires collected, calculations for samples sizes targeted, inclusion criteria and exclusion criteria

2- Which type of the questions were used in the survey?

Results

1- The abbreviation JoD is not used. The correct one is JD (Jordanian Dinars). Please change it everywhere

2- The Figures shouldn’t have title inside the figure. The figure caption should be enough. Additionally more details should be mentioned in the figure’s captions to make it clearer

Discussion and conclusion

This section is good with good comparisons to other studies.. I suggest to shorten the conclusion a bit

6. PLOS authors have the option to publish the peer review history of their article (what does this mean?). If published, this will include your full peer review and any attached files.

Reviewer #1: **Yes: **Dr. Rana Abutaima

Reviewer #2: No

---

## [Author Response · Author response to Decision Letter 0]

3 Jul 2024

Response to Reviewers

Additional Editor Comments:

Many thanks for your valuable comments.

1. Title: Shorten the title to no more than 15 words for conciseness, and please remove the abbreviation.

Response: The title was shorted to: “Investigation of the Knowledge, Attitudes, and Perceptions Regarding the Utilization of Rosemary among the Population in Jordan”.

2. Abstract: Your email is to validate questionnaire and assess the knowledge. Please added aims accordingly. Include details about your data collection methods such as whether you used random, snowball, or convenience sampling. Moreover, add the time of data collection. Construct validity why is not included in your study?

Response: 

- The aim of the study is to assess the knowledge, attitudes, and perceptions of the general population regarding the use of rosemary and its oil. While developing the questionnaire, previously published similar questionnaires were used to extract questions and then the face and content validity as well as measuring Cronbach's alpha was enough to ensure its validity. 

- The used questionnaires were referenced within the manuscript.

- Cronbach's alpha was also included in the “Materials and Methods” section.

- The following sentence was added: “Internal consistency reliability was tested by Cronbach's alpha coefficient, which resulted in a coefficient of 0.76”.

- The following paragraph was added: “A snowball sampling method was employed. As initial potential participants were recruited, and then they were asked to refer other potential participants who met the study's inclusion and were willing to complete the survey. The process continued until the reached sample size was sufficient and representative to meet the study's objectives.”

3. Introduction: References in Brackets. Add paragraph about scales used to measure knowledge and why the selected scale is important. Add the aim of validating a questionnaire. Add what the problem or useful of using Rosmarinus among Jordanians.

Method Section: Provide more information on how you conducted the online survey, ensuring it is clear how participants from Jordan were able to respond. How you reached 407 of participants. Is any outliers in incomplete surveys.

Response: 

- The referencing style was edited as required.

- The following paragraphs were added to the materials and methods section: 

“A good degree of knowledge was defined in this study as having at least 50% of the total right answers [26]. Additionally, a higher score in each category denotes greater understanding.”

“'Yes' and 'No' dichotomous questions were chosen for the knowledge section. This format was selected due to its ability to precisely target the specific knowledge area relevant to the current research, which may not be fully covered in the existing published literature [27].”

- Since the questionnaire was not developed from scratch, but rather from previously published questionnaires, face and content validity as well as getting Cronbach's alpha results were found to be enough to ensure its reliability and consistency.

- The following sentence was added: “Internal consistency reliability was tested by Cronbach's alpha coefficient, which resulted in a coefficient of 0.76”.

- A paragraph was added to the introduction: “Rosemary is cultivated in Jordan making it easily accessible and available to the general population [15, 16]. It was reported in several studies that Jordanians use rosemary for various health conditions [17, 18].”

- The following paragraph was added: “Internal consistency reliability was tested by Cronbach's alpha coefficient, which resulted in a coefficient of 0.76.”

- The following paragraph was added to the Materials and Methods section: 

“Sample size

Currently, the number of Jordanian citizens living in Jordan is 11.516 million [27]. Based on that, calculating the sample size using a margin of error of 5%, confidence level of 95%, and response distribution of 50%, a minimum sample size of 385 is needed. The number of participants in this study was 407 [28].”

- The following sentence was added to the materials and methods section: “Eligible participants were any interested Jordanian citizens living in Jordan. The questionnaire was distributed using WhatsApp® and other social media platforms such as Facebook®, Twitter® and LinkedIn®.”

- The following paragraph was added to the results section: “A total of 409 potential participants opened the study's survey. Out of these, two participants chose the ‘disagree’ button, indicating their refusal to participate. As a result, 407 participants were included in the analysis, yielding a response rate of 99.51%.”

A. Questionnaire development: Where are the references regarding development? Consider adding construct validity measures such as correlations, KMO, Bartlett's test of sphericity, and total variation in addition to CVR ratios (content validity).

Response: 

- The used questionnaires were referenced within the manuscript.

- As mentioned before, Since the questionnaire was not developed from scratch, but rather from previously published questionnaires, the questions were adopted and the name of the herb(s) was changed to rosemary, in addition, the common health benefits of it were also extracted from the literature and incorporated into the questions. Thus, face and content validity as well as getting Cronbach's alpha results were found to be enough to ensure its reliability and consistency.

B. Typos: Carefully review and correct any typos present in the manuscript.

Response: The manuscript was reviewed, and typos were corrected.

C. Discussion: Begin the discussion section with a focus on your main findings rather than repeating the study's aims.

International Comparisons: Compare your results with findings from international studies to highlight key differences.

Response: 

- The following sentence were used to start the discussion section: “This is the first study to focus on the use of one of the most common natural products available in Jordan, rosemary.” Additionally, the following sentence was also added to the first paragraph in the discussion section: “knowledge, where age was found to be the only significant variable that affected the rosemary knowledge score.”

- No exact previous studies were found in the literature, so the results were compared with similar ones conducted in the United States, the United Kingdom, Australia and Germany Ukraine and Korea. 

- The following paragraphs were added: “Moreover, in a study conducted in Morocco targeting patients with kidney diseases revealed that that most used herbal medicine was Rosmarinus officinalis [32].”

- “In Türkiye, friends were found to be the main source of information for herbal medicines users [20]. Additionally, in a survey conducted in central region of Saudi Arabia, the most common influences for using herbal medicine were found to be family and friends followed by the internet [19].”

D. References: Double-check and ensure the accuracy of your references.

Response: Done.

..................................................................................................................................................

Reviewer #1: 

Thank you for taking the time to review the submitted article and providing us with valuable comments.

Reviewer comments:

Abstract

• Write the unit for the age.

Response: added.

• 70. Should be written in percentage form

Response: added.

• The introduction writing sounds more literary than scientific. It needs to focus scientifically on the uses, active ingredients and how they proved therapeutic effect and significance.

Response: The introduction was re-written as advised. 

Methodology:

• It is well-written but there is a redundancy in the sentence “to proceed with the study study's survey” needs to be edited.

Response: corrected.

• How the survey was disseminated to the public? You need to mention this in the methodology section

Response: The following sentence was added to the materials and methods section: “The questionnaire was distributed using WhatsApp® and other social media platforms such as Facebook®, Twitter® and LinkedIn®.”

Results:

• It is a good practice to mention if there were incomplete responses or if there was any response that was excluded from the analysis.

Response: The following paragraph was added to the results section: “A total of 409 potential participants opened the study's survey. Out of these, two participants chose the ‘disagree’ button, indicating their refusal to participate. As a result, 407 participants were included in the analysis, yielding a response rate of 99.51%.”

• Define abbreviations at the first instance (e.g. JoD)

Response: corrected and mentioned for the first time within the text.

• Figure 2 and 3: define the y-axis title

Response: Done

• Few grammatical issues such as “its’”

Response: revised and corrected.

• I think that some statements containing negation such as “Rosemary cannot be used as a natural aroma” were leading.

Response: This was done as advice obtained from an expert while developing the survey. As one way to make sure that the participants are actually reading all the statements and then answering it accordingly is by using positive and negative indicators for balance. As, people who agree with the positive statements should disagree with the negative ones, and vice versa. That's how we can be confident that the questions are clear, and the responses are reliable.

.............................................................................................................................................................

Reviewer #2: This study investigates the Jordanians’ attitude, practice and knowledge towards rosemary and rosemary oil. I suggest accepting it for publication but after amending it based on the following comments

Thank you for your time in reviewing the submitted article and for providing us with enriching comments. 

Title

Although in the title the authors mentioned “Knowledge” first, they left everything about it in the M&M and Results to the end. I don’t agree to that because first you have to investigate about the knowledge then the attitude and the practice. Therefore, I suggest that the authors arrange the manuscript to talk first about the knowledge then the other components. In this way there will be synchronization between the title and the different parts of the manuscript. I strongly encourage sending the manuscript to English editing.

Response: The manuscript was arranged as advised and was reviewed to correct linguistic errors.

Abstract:

1- The last sentence in the background has to be reformulated

Response: Done.

2- In the Methods part, it is better to mention the sample number

Response: The following sentence was added: “A cross-sectional study was conducted targeting at least 385 participants via social media platforms.”

3- In the method section, you have to arrange the sentence talking about the what this paper is assessing. You should start with knowledge then practice and attitude

Response: Done.

4- In the methods, it would be important to mention which geographic parts of Jordan where covered

Response: It was an online based questionnaire, so adults at any part of Jordan can respond. 

5- In the results part: add % for the last number of “More than half reported using it as inhalation (55.0%), whereas 77.6% administered it orally, and 70.9 applied it topically.”

Introduction

Response: The % sign was added and some paragraphs in the abstract were re-written.

Introduction

1- It would be more informative if you mention about the percentage of the population in geographic regions other than in the Arab world in the first paragraph. This would be useful to see the differences.

Response: The introduction was re-written as advised. 

2- You need to add reference for this sentence “Rosemary's historical significance is far-reaching and diverse: from being revered in Ancient Greece as a memory-enhancing agent to gracing the burial rituals of Egyptian pharaohs while also being utilized in the embalming of corpses.”

Response: The introduction was re-written as advised. 

3- I think the introduction needs to have few sentences more mentioning if similar study was conducted previously in jordan and in other countries or not. This should lead to the concluding sentence about the importance of this study, at the end of the Introduction section

Response: The introduction was re-written as advised. 

M&M

1- This section is missing information about the geographic areas targeted, number of questionnaires collected, calculations for samples sizes targeted, inclusion criteria and exclusion criteria

Response: The following paragraphs were added to the materials and methods section:

- “Eligible participants were any interested Jordanian citizens living in Jordan. The questionnaire was distributed using WhatsApp® and other social media platforms such as Facebook®, Twitter® and LinkedIn®.”

- “Sample size: Currently, the number of Jordanian citizens living in Jordan is 11.516 million [27]. Based on that, calculating the sample size using a margin of error of 5%, confidence level of 95%, and response distribution of 50%, a minimum sample size of 385 is needed. The number of participants in this study was 407 [28].”

- The following paragraph was added to the results section: “A total of 409 potential participants opened the study's survey. Out of these, two participants chose the ‘disagree’ button, indicating their refusal to participate. As a result, 407 participants were included in the analysis, yielding a response rate of 99.51%.”

2- Which type of the questions were used in the survey?

Response: The questionnaire is provided as an appendix (S1 Appendix). 

Results

1- The abbreviation JoD is not used. The correct one is JD (Jordanian Dinars). Please change it everywhere

Response: Done.

2- The Figures shouldn’t have title inside the figure. The figure caption should be enough. Additionally more details should be mentioned in the figure’s captions to make it clearer

Response: Done, the title inside the figure was removed and the figure title was changed.

Discussion and conclusion

This section is good with good comparisons to other studies. I suggest to shorten the conclusion a bit.

Response: The conclusion was shortened as advised.

---

## [Editor Report · Decision Letter 1]

9 Jul 2024

Investigation of the Knowledge, Attitudes, and Perceptions Regarding the Utilization of Rosemary among the Population in Jordan

PONE-D-24-15316R1

Dear Dr.

We’re pleased to inform you that your manuscript has been judged scientifically suitable for publication and will be formally accepted for publication once it meets all outstanding technical requirements.

Kind regards,

Othman A. Alfuqaha, Ph.D.

Academic Editor

PLOS ONE

Additional Editor Comments (optional):

Congratulations on your valuable work.
---

## [Editor Report · Acceptance letter]

11 Jul 2024

PONE-D-24-15316R1 

PLOS ONE

Dear Dr. Thiab, 

I'm pleased to inform you that your manuscript has been deemed suitable for publication in PLOS ONE. Congratulations! Your manuscript is now being handed over to our production team.

Kind regards, 

on behalf of

Dr. Othman A. Alfuqaha 

Academic Editor

PLOS ONE